# Association of Medicare Program Type with Health Care Access, Utilization, and Affordability among Cancer Survivors

**DOI:** 10.3390/cancers15153964

**Published:** 2023-08-04

**Authors:** Faraz I. Jafri, Vishal R. Patel, Jianhui Xu, Daniel Polsky, Arjun Gupta, Syed Mohammed Qasim Hussaini

**Affiliations:** 1Dell Medical School, The University of Texas at Austin, Austin, TX 78712, USA; vishpatel97@utexas.edu; 2Department of Health Policy & Management, Johns Hopkins Bloomberg School of Public Health, Baltimore, MD 21205, USA; jxu123@jhu.edu (J.X.); polsky@jhu.edu (D.P.); 3Division of General Internal Medicine, Johns Hopkins Medicine, Baltimore, MD 21224, USA; 4Carey Business School, Johns Hopkins University, Baltimore, MD 21218, USA; 5Masonic Cancer Center, University of Minnesota, Minneapolis, MN 55455, USA; arjgupta@umn.edu; 6Sidney Kimmel Comprehensive Cancer Center, Johns Hopkins Medicine, Baltimore, MD 21224, USA; shussa11@jh.edu

**Keywords:** medicare advantage, traditional medicare, cancer survivors, affordability

## Abstract

**Simple Summary:**

This research study aims to understand whether the Medicare Advantage program, which provides healthcare to a significant portion of Medicare beneficiaries, including cancer survivors, is effectively improving access to care, healthcare outcomes, and affordability. The study highlights the need for the ongoing evaluation and re-evaluation of the role of Medicare Advantage in promoting equity among beneficiaries with cancer. It provides valuable insights for policymakers and healthcare professionals involved in decision-making processes related to Medicare Advantage and its impact on healthcare access, affordability, and financial well-being for cancer survivors. This research informs discussions on potential improvements in the Medicare program to ensure that the needs of cancer survivors are adequately addressed.

**Abstract:**

Background: The Medicare Advantage program provides care to nearly half of Medicare beneficiaries, including a rapidly growing population of cancer survivors. Despite its increased adoption, it is still unknown whether or not the program improves healthcare access, outcomes, and affordability for cancer survivors. Methods: We performed a cross-sectional study of Medicare beneficiaries aged ≥ 65 years with a self-reported history of cancer from the 2019 National Health Interview Survey. We used multivariable logistic regression to evaluate the association between Medicare program type (Medicare Advantage vs. traditional Medicare) and measures of healthcare access, acute care utilization, and affordability. Results: We identified 4451 beneficiaries with a history of cancer, corresponding to 26.6 million weighted cancer survivors in 2019. Of the beneficiaries, 35.8% were enrolled in Medicare Advantage, whereas 64.2% were enrolled in traditional Medicare. The age, sex, racial and ethnic composition, household income, primary site of cancer, and comorbidity burden of Medicare Advantage and traditional Medicare beneficiaries were similar. In the adjusted analysis, there were no differences in healthcare access or acute care utilization between traditional Medicare and Medicare Advantage beneficiaries. However, cancer survivors enrolled in Medicare Advantage were more likely to worry about (34.3% vs. 29.4%; aOR, 1.3 (95% CI, 1.1–1.5)) or have problems paying (13.6% vs. 11.1%; aOR, 1.4 (95% CI, 1.1–1.8)) medical bills. Conclusions: We found no evidence that Medicare Advantage beneficiaries with cancer had better healthcare access, affordability, or acute care utilization than traditional Medicare beneficiaries did. Furthermore, Medicare Advantage beneficiaries were more likely to report financial strain and have difficulty paying for their medical bills than were those with traditional Medicare. Despite the generous benefits and attractive incentives, Medicare Advantage plans may not be more cost-effective than traditional Medicare is for cancer survivors. Our study informs ongoing congressional deliberations to re-evaluate the role of Medicare Advantage in promoting equity among beneficiaries with cancer.

## 1. Introduction

Patients with cancer are disproportionately impacted by financial strain and often face barriers to accessing care, both of which have been linked to poorer health outcomes [1,2,3,4,5,6]. The expensiveness of care and the need for the frequent monitoring of cancer survivors create a unique burden that often outlasts their initial diagnosis and treatment. Thus, increasing attention is being paid to reducing the burden of cancer care to improve outcomes for survivors. 

One such avenue to improving healthcare access and affordability is through healthcare insurance design [7]. While traditional Medicare is often supplemented with private insurance through Medigap, Medicare Advantage provides beneficiaries with all of its Part A and Part B benefits through a contract between private insurance companies and Medicare [8]. Medicare Advantage includes additional benefits and services not covered by traditional Medicare, such as supplemental dental and vision coverage, and better care co-ordination [9]. Further, there is a limit on out-of-pocket costs each year for covered services, an offer not extended to those with traditional Medicare. Most Medicare Advantage plans are funded by risk-adjusted, fixed, per-person payments for beneficiaries, which creates an incentive to be cost effective when delivering care. While this incentive could lead to a larger emphasis on value-based and preventative care, it could also lead to the rationing of healthcare resources and possible avoidance of necessary care [10]. Further, Medicare Advantage plans can be a valuable tool for managing the high costs of cancer treatment, but may include more restrictions on the providers and services that are covered, and may require higher premiums than traditional Medicare does [11]. 

While the prior literature has shown that Medicare Advantage is associated with increased affordability in certain patient subgroups, there is conflicting evidence that it improves quality of care when compared to traditional Medicare [12,13,14]. Furthermore, little is known about whether or not differences in healthcare affordability, utilization, and access exist within cancer survivors. In this study, we explored the variability in preventative and acute care utilization, and the barriers to affordability in cancer survivors on traditional Medicare or Medicare Advantage.

## 2. Methods

### 2.1. Study Population

We included all adults aged ≥ 65 years in the 2019 National Health Interview Survey (NHIS) who were enrolled in Medicare and had a self-reported history of cancer. The NHIS is a nationally representative cross-sectional survey administered by the Centers for Disease Control each year and is the premiere source for patient-reported health outcomes in the United States [15]. Using the NHIS, we extracted self-reported Medicare plans (Medicare Advantage vs. traditional Medicare) for each beneficiary. We also extracted self-reported patient demographic information, including age, sex, race, ethnicity, annual household income, metropolitan status, region, and patient clinical information, including the primary site of cancer and presence of chronic comorbidities (i.e., diabetes, hyperlipidemia, hypertension, coronary heart disease, stroke, chronic obstructive pulmonary disease, asthma, dementia, and depression). 

### 2.2. Outcomes

Outcomes comprised several self-reported measures in the following areas: (1) healthcare access, (2) acute care utilization, and (3) affordability. Measures of healthcare access included access to (a) a usual source of care, (b) a physician visit in the previous year, and the routine monitoring of (c) cholesterol, (d) blood pressure, and (e) blood glucose in the previous year. Measures of acute care utilization included (a) ≥2 urgent care visits in the previous year, (b) at least one emergency department visit in the prior year, and (c) at least one overnight hospital stay in the prior year. Measures of healthcare affordability included (a) delaying care to save money in the previous year, (b) a general worry about paying medical bills in the previous year, and (c) difficulty in paying or an inability to pay medical bills in the previous year. The affordability of prescription medication was assessed as (d) delaying filling in prescription medication or (e) skipping doses of prescription medication to save money in the previous year.

### 2.3. Statistical Analysis

We compared the baseline characteristics of adults enrolled in either Medicare Advantage or traditional Medicare using the two-sample t-test and chi-squared test. Next, we computed the unadjusted rates of each outcome for both Medicare groups and derived unadjusted odds ratios (ORs) using empty logistic regression models. Finally, we adjusted the models for patient demographic factors (age, sex, race, ethnicity, and income), comorbidities (hypertension, hyperlipidemia, diabetes, coronary heart disease, stroke, chronic obstructive pulmonary disorder, dementia, asthma, and depression), primary cancer site, and geography (region and rurality). We weighted the estimates to account for NHIS’s complex survey design. Because the data source was publicly available and de-identified, the study was exempt from institutional review board approval and the need for informed consent, in accordance with 45 CFR §46. Two-sided *p*-values (α < 0.05) were applied to the univariate tests. STATA (v17; StataCorp) was used for all analyses.

## 3. Results

We identified 4451 participants, resulting in a weighted cohort of 26,619,345 beneficiaries with prior cancer diagnosis. Of the beneficiaries, 9,521,181 (35.8%) were enrolled in Medicare Advantage and 17,098,164 (64.2%) were enrolled in traditional Medicare. Age, sex, race and ethnicity, income, primary cancer site, and comorbidity burden were similar for survivors enrolled in Medicare Advantage and traditional Medicare (Table 1). However, Medicare Advantage beneficiaries were more likely to live in urban settings (86.2% vs. 78.0%, *p* < 0.01) and the western region of the United States (23.9% vs. 17.9%, *p* < 0.01). 

### 3.1. Healthcare Access

Compared to cancer survivors enrolled in traditional Medicare, those enrolled in Medicare Advantage were equally likely to have a usual place of care (95.5% vs. 94.8%; adjusted odds ratio (aOR), 1.2 (95% CI, 0.8–1.8)) and a routine physician visit in the last year (98.2% vs. 97.7%; aOR, 1.3 (95% CI, 0.8–2.2)) (Table 2). Survivors with Medicare Advantage and traditional Medicare were also equally likely to have undergone a recent cholesterol screening (93% for Medicare Advantage vs. 94.4% for traditional Medicare; aOR, 0.8 (95% CI, 0.5–1.3)), and had their blood glucose (90.7% vs. 91.2%; aOR, 1 (95% CI, 0.7–1.5)) or blood pressure evaluated (99.3% vs. 98.6%; aOR, 1.9 (95% CI, 0.6–6.2)) in the last year. 

### 3.2. Acute Care Utilization

Compared to survivors enrolled in traditional Medicare, those enrolled in Medicare Advantage were similarly likely to have had an emergency department visit in the prior year (28.3% for Medicare Advantage vs. 29.6%; aOR, 0.9 (95% CI, 0.8–1.1)), two or more urgent care visits in the prior year (9.9% vs. 11.1%; aOR, 0.9 (95% CI, 0.7–1.2)), and overnight hospitalizations in the prior year (20.4% vs. 20.5%; aOR, 1 (95% CI, 0.8–1.2)) (Table 3).

### 3.3. Healthcare Affordability

Compared to cancer survivors with traditional Medicare, those with Medicare Advantage were significantly more likely to report worrying about medical bills (34.3% for Medicare Advantage vs. 29.4% for traditional Medicare; aOR, 1.3 (95% CI, 1.1–1.5)) and had trouble paying medical bills (13.6% vs. 11.1%; aOR, 1.4 (95% CI, 1.1–1.8)) in the previous year (Table 4). Both groups were similarly likely to delay filling prescriptions (5.5% vs. 5%; aOR, 1.1 (95% CI, 0.8–1.5)) and skip doses of medications (4.3% vs. 3.2%; aOR, 1.5 (95% CI, 1–2.3)) to save money. 

## 4. Discussion

In this nationally representative study of cancer survivors on Medicare, access to and use of health care did not significantly differ between beneficiaries on Medicare Advantage or traditional Medicare. Further, Medicare Advantage beneficiaries were more likely to report financial strain and have difficulty paying for their medical bills than those with traditional Medicare. 

When Medicare Advantage was launched in 2003, agencies expected improved affordability and widened coverage as the program provided protection from extreme out-of-pocket expenses and added supplemental benefits. Previous studies on Medicare Advantage have demonstrated reduced expenses but have varied results regarding health quality, access, and outcomes [10,12,13,14,16,17]. In our study, we noted a lack of improvement in utility and accessibility among Medicare Advantage plans for cancer survivors, which raises important questions about the value and cost-effectiveness of the extra benefits offered to beneficiaries. One possible cause of these similarities may be the numerous upstream social determinants of health that were not assessed in this study and could affect both groups, regardless of medical insurance [12]. The similarities indicate a need for a re-evaluation of the program to ensure that Medicare beneficiaries, especially cancer survivors, are receiving the best possible care at an affordable cost and that taxpayer dollars are being used effectively.

There are several possible reasons beneficiaries with Medicare Advantage are more likely to have difficulty paying for their medical bills. One reason may be that Medicare Advantage plans may engage in “cream skimming” under capitated payments, making the plans less appealing to cancer survivors by less generously covering the services they use [18]. In comparison, over 80% of traditional Medicare beneficiaries have some form of supplemental coverage that reduces their out-of-pocket burden through Medigap policies, employer-sponsored retiree health benefits, or Medicaid [19]. Medicare Advantage beneficiaries are protected by a maximum out-of-pocket limit, which varies by plan and provider. However, certain plans (specifically HMOs) offer a narrower network of providers, which could limit beneficiaries’ access to care and potentially result in higher out of pocket costs for those who need to seek care outside of their network. These findings suggest that high-need Medicare Advantage beneficiaries may be more vulnerable to financial strain owing to their healthcare expenses.

Taken together, our findings suggest that relative to traditional Medicare, Medicare Advantage has not provided significant improvements in healthcare access or utilization for cancer survivors, and it may even reduce affordability. This is particularly concerning given that enrollment among adults with low incomes and that among Black and Hispanic adults has increased rapidly within the Medicare Advantage program [10,20]. The findings are also consistent with those of prior studies that Medicare Advantage beneficiaries with high healthcare needs are more likely to switch from Medicare Advantage to traditional Medicare [21,22]. Given the rapid growth of Medicare Advantage, it is crucial to explore avenues for improving beneficiary satisfaction. Recent actions towards this goal include regulations to strengthen the risk adjustment program and thus reduce plans’ cream skimming incentives [23]. Policymakers can also play a role in enhancing consumer protection against high cost-sharing and ensure that plans’ networks provide beneficiaries with adequate access to healthcare providers.

Several limitations should be considered when interpreting the results. First, our data lacked any adjustments made for clinical variables, such as cancer staging or treatment details associated with Medicare program type, accessibility, affordability, and utilization, which were not captured in our analysis. The NHIS is not a clinical database and thus has limited clinical information regarding the reception of cancer-directed treatment or the timing of such treatment. It was specifically selected because it captures extensive demographic data and a broad spectrum of access and utilization measures that are not found in many other clinical or claims-based data. Second, differences in enrollment by state-based policies or location of residence were limited to broad regional differences because only information on the census region was available for each participant. Third, our data did not include information on secondary coverage such as Medigap, which may have affected beneficiaries’ cost sharing. Fourth, we could not account for the variability in coverage within Medicare Advantage plans depending on the insurer, which means that some plans may perform better or worse than others. However, resolving within-plan variation was not mentioned as part of the specific aims of the current study, which was only intended to resolve between-plan differences. Finally, it is important to note that although the non-significance of our results may be attributed to the smaller sample size, we used a nationally representative sample of cancer survivors for the most recently available survey year containing all outcomes of interest. Finally, the survey data used in the study are subject to response bias, recall bias, and survivorship bias, which may affect the accuracy of the results.

## 5. Conclusions

In this nationally representative study of Medicare beneficiaries in 2019, we found that there were no meaningful differences in healthcare access, affordability, and utilization between cancer survivors enrolled in Medicare Advantage and traditional Medicare. Despite offering more comprehensive benefits and incentives to improve the coordination and management of health, our findings indicate that Medicare Advantage plans may not result in substantial advancements in key health-related outcomes for cancer survivors. These results call into question the effectiveness of Medicare Advantages in improving health access and affordability within the Medicare program.

## Figures and Tables

**Table 1 cancers-15-03964-t001:** Characteristics of Medicare beneficiaries with cancer, by type of Medicare plan.

Characteristic	Beneficiaries, Weighted %		*p* Value
	Traditional Medicare (n = 17,098,164) ^a^	Medicare Advantage (n = 9,521,181) ^a^	
Age, mean (SD), y	76.2 (49.7)	75.8 (47.7)	0.85
**Sex**			
Male	46.0	43.3	0.12
Female	54.0	56.7	
**Race and ethnicity**			
White	82.5	79.1	0.21
Black	8.1	9.6	
Asian	2.0	2.5	
other	1.7	2.7	
Hispanic	5.6	6.2	
**Income, (% of the FPL)**			
<100	8.3	7.5	0.68
100–199	21.3	21.8	
200–399	32.7	34.6	
≥400	37.4	35.9	
Unknown	0.3	0.2	
**Metropolitan status**			
Non-urban	22.0	13.8	<0.01
Urban	78.0	86.2	
**Region**			
Northeast	20.4	15.9	<0.01
Midwest	22.0	21.8	
South	39.7	38.4	
West	17.9	23.9	
**Primary site ^b^**			
Breast	23.2	25.6	0.2
Gastrointestinal	8.4	8.1	
Gynecologic ^c^	10.4	10.9	
Head & Neck	5.1	5.4	
Hepatopancreatobiliary	1.6	1.2	
Leukemia or lymphoma	4.5	6.2	
Lung or bronchus	4.3	4.6	
Melanoma	5.9	6.3	
Other	11.0	8.5	
Urologic ^d^	25.5	23.2	
**Co-morbidities**			
Diabetes	22.6	22.3	0.84
Hyperlipidemia	56.3	54.3	0.25
Hypertension	65.3	65.1	0.93
Coronary heart disease	17.5	17.5	0.99
Stroke	9.6	9.5	0.97
COPD	13.6	14.5	0.42
Asthma	13.1	14.4	0.31
Dementia	4.5	3.8	0.42
Depression	18.0	20.2	0.15

Abbreviations: FPL, federal poverty limit; COPD, chronic obstructive pulmonary disease. ^a^ Sample sizes represent nationally representative weighted estimates from the 2019–2021 National Health Interview Survey (unweighted sample sizes were n = 2871 for Traditional Medicare and n = 1580 for Medicare Advantage). ^b^ The primary site of the most recently diagnosed cancer is shown for individuals who reported multiple prior cancers. ^c^ Cancer of the cervix, ovary, or uterus. ^d^ Cancer of the bladder, kidney, or prostate.

**Table 2 cancers-15-03964-t002:** Health care access among beneficiaries with cancer, by type of Medicare plan.

Outcome	Beneficiaries, % ^a^		Unadjusted Odds Ratio (95% CI)	*p* Value	Adjusted Odds Ratio (95% CI) ^b^	*p* Value
	**Medicare Advantage**	**Traditional Medicare**				
Has a usual source of care	95.5	94.8	1.2 (0.8–1.7)	0.35	1.2 (0.8–1.8)	0.29
Had a doctor visit in the last year	98.2	97.7	1.3 (0.8–2)	0.35	1.3 (0.8–2.2)	0.33
Met aerobic physical activity recommendations	31.8	32.3	1.0 (0.7–1.3)	0.89	1.0 (0.7–1.3)	0.78
Had blood pressure evaluated in the last year	99.3	98.6	2.1 (0.7–6.6)	0.21	1.9 (0.6–6.2)	0.27
Had blood glucose evaluated in the last year	90.7	91.2	0.9 (0.7–1.3)	0.74	1.0 (0.7–1.5)	0.87
Had cholesterol evaluated in the last year	93	94.4	0.8 (0.5–1.2)	0.29	0.8 (0.5–1.3)	0.31

Odds ratios are presented with traditional Medicare as the reference group. ^a^ Percent of beneficiaries answering yes. ^b^ Adjusted for age, sex, education, race and ethnicity, annual income, primary site of cancer, number of chronic diseases, region of residence, metropolitan status, and survey year.

**Table 3 cancers-15-03964-t003:** Acute care utilization among beneficiaries with cancer, by type of Medicare plan.

Outcome	Beneficiaries, % ^a^		Unadjusted Odds Ratio (95% CI)	*p* Value	Adjusted Odds Ratio (95% CI) ^b^	*p* Value
	Medicare Advantage	Traditional Medicare				
Visited an urgent care clinic ≥ 2x in the last year	9.9	11.1	0.9 (0.7–1.1)	0.31	0.9 (0.7–1.2)	0.37
Visited an emergency department in the last year	28.3	29.6	0.9 (0.8–1.1)	0.47	0.9 (0.8–1.1)	0.41
Had an overnight hospitalization in the last year	20.4	20.5	1.0 (0.8–1.2)	0.92	1.0 (0.8–1.2)	0.96

Odds ratios are presented with traditional Medicare as the reference group. ^a^ Percent of beneficiaries answering yes. ^b^ Adjusted for age, sex, education, race and ethnicity, annual income, primary site of cancer, number of chronic diseases, region of residence, metropolitan status, and survey year.

**Table 4 cancers-15-03964-t004:** Health care affordability among beneficiaries with cancer, by type of Medicare plan.

Outcome	Beneficiaries, % ^a^		Unadjusted Odds Ratio (95% CI)	*p* Value	Adjusted Odds Ratio (95% CI) ^b^	*p* Value
	Medicare Advantage	Traditional Medicare				
Worried about paying medical bills in last year	34.3	29.4	1.3 (1.1–1.5)	0.01	1.3 (1.1–1.5)	0.01
Had problems paying or inability to pay medical bills in last year	13.6	11.1	1.3 (1–1.6)	0.06	1.4 (1.1–1.8)	0.02
Delayed medical care to save money in the last year	4.3	3.9	1.1 (0.7–1.7)	0.62	1.3 (0.8–1.9)	0.32
Delayed filling medications to save money in the last year	5.5	5.0	1.1 (0.8–1.6)	0.51	1.1 (0.8–1.5)	0.66
Skipped prescription medications to save money in last year	4.3	3.2	1.4 (0.9–2)	0.12	1.5 (1–2.3)	0.06

Odds ratios are presented with traditional Medicare as the reference group. ^a^ Percent of beneficiaries answering yes. ^b^ Adjusted for age, sex, education, race and ethnicity, annual income, primary site of cancer, number of chronic diseases, region of residence, metropolitan status, and survey year.

## Data Availability

Publicly available dataset can be found at https://nhis.ipums.org/nhis/.

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
