# Peer review of "Association of Medicare Program Type with Health Care Access, Utilization, and Affordability among Cancer Survivors"

_cancers, 2023, doi:10.3390/cancers15153964_

Round 1
Reviewer 1 Report
I enjoyed reading this work by Jafri and colleagues. They utilized the NHIS to evaluate access to care and care affordability among cancer survivors with Medicare Advantage and traditional Medicare. They found that access measures were comparable between Medicare Advantage and traditional Medicare enrollees, but measures of care affordability were worse among Medicare enrollees. These are very important data that can inform consumers and policymakers about the Medicare program and potential disadvantages of current Medicare Advantage plans for cancer survivors. I felt the data and statistical analysis were appropriate for the study, and the writing was clear. I also thought the authors did a nice job outlining the major study limitations. I have a couple comments for the authors’ consideration:
I believe some (small proportion of) patients in the NHIS are currently undergoing cancer treatment, and some may have undergone cancer treatment more recently. Do the authors have any information on these baseline factors? I would expect that these factors could affect insurance decisions as well as care affordability, potentially confounding findings.
Additional minor comments:
The authors state that Medicare advantage has been shown to improve affordability and quality of care (introduction lines 73-74, discussion lines 184-186). However, the (limited) data involving patients with cancer, at least with which I am familiar, suggests otherwise (for example, see PMID: 36356283, which I didn’t see in the references list). I would suggest tempering the language utilized.
While I agree with the authors’ recommendations in lines 214-221, I don’t think they are supported by data. While I think these points can still be discussed, I would suggest tempering the language used here.
While this might be a product of the PDF conversion, each of the Tables has abnormal bolding of part of the first line under the headers. Please fix.
Reviewer 2 Report
Thank you for the opportunity to review this interesting paper focused on differences in quality, access, and affordability of health care for cancer survivors enrolled in the two types of Medicare - Medicare Advantage (MA) and traditional Medicare (TM). This paper provides important results that contribute to the evidence base regarding the performance of MA plans as compared to TM.
In general, I found the study design and results to be believable and largely compelling. However, I do have some concerns that I believe the authors should address before the paper is considered for publication.
These concerns include:
1. The use of the NHIS survey data is an interesting approach, but the manner by which the authors convert the ~4,500 beneficiary respondents with previous cancer diagnosis to the total of 26 million beneficiaries is unclear. Further, I'm not sure this makes sense - based on a quick Google search, about 1 million Medicare beneficiaries per year are diagnosed with cancer. Even if all of these beneficiaries survived, it would take decades to achieve 26 million total with a previous cancer diagnosis. I would recommend the authors revisit these analyses and methods and either update the total weighted estimates to more closely reflect reality or better explain how they achieved these numbers to begin with.
2. The authors focus on the one statistically significant difference in survey reported outcomes, that being worrying about affordability of care. In the Discussion, the authors acknowledge an important piece, that being the presence of secondary insurance coverage - most importantly, Medigap coverage for those enrolled in TM. This coverage most likely reduces concerns by TM enrollees regarding affordability of care, much more so compared to MA enrollees, who do not have access to this type of coverage. I would recommend mentioning the Medigap option much earlier on in the paper and considering whether to control for beneficiaries having this type of secondary coverage in your regression models. If the survey data do not include information on secondary coverage, this should likely be noted as a limitation of these analyses.
3. While the focus on cancer survivors is compelling, I am left wondering who these survivors are and about the likely broad range of care they require upon entering remission with cancer. Some cancers likely require much less care after initial treatment compared to others (skin cancer comes to mind as an excellent example). Without being able to distinguish between the types of cancer being disclosed, this raises questions about the severity of the cancers and whether or not more beneficiaries with more severe cancers were enrolled in TM as opposed to MA (as an example).
4. Related to #3, I also wonder about the timing of cancer diagnosis and survey responses. Beneficiaries who recovered from cancer 5 or more years before likely look very different from beneficiaries who were diagnosed a year ago and may even still be in treatment for their cancer. It will be important for the authors to explain how they distinguished across cancer timing and the survey reporting approaches used.
5. A more minor comment, I would recommend the authors revisit the description of the MA Program in the Introduction. My understanding of the introduction of MA was that Congress wanted to offer beneficiaries an alternative option to TM. There was likely talk that private insurers might be able to offer better care coordination, better quality, and lower costs for beneficiaries, but I'm not sure those were all explicitly stated goals of MA - also not that MA did not technically exist until 2003. I think that more emphasis/discussion about the protective nature of the max OOP limits in MA, which do not exist in TM, is also warranted - while often only applied to receipt of care that is in-network, this is a huge advance beyond TM and a big factor in affordability for beneficiaries who choose MA compared to TM.
The quality of the English language is very high, there are just a few typos here and there that could benefit from a proofread before final publication.
